# Transcriptomic Analysis Reveals the Mechanism of Lignin Biosynthesis in Fresh-Cut Cucumber

Yi Wang [1], Ning Yuan [1], Yuge Guan [2], Chen Chen [1],* and Wenzhong Hu [1],*

1   College of Life Science, Dalian Minzu University, Dalian 116600, China; w13936057663@163.com (Y.W.); yn2822852617@163.com (N.Y.)
2   School of Food and Health, Zhejiang Agricultural and Forestry University, Hangzhou 311300, China; gyg@zafu.edu.cn
*   Correspondence: chenchen@dlnu.edu.cn (C.C.); huwenzhongtd@sina.com (W.H.)

**Abstract:** When subjected to a certain degree of mechanical damages, a systematic responsive mechanism of fresh-cut cucumber is activated. Among them, the lignin produced in the secondary metabolism will make the fresh-cut cucumber lignified, which will increase the hardness and whiten the cutting surface of the fresh-cut cucumber, seriously affecting the taste and appearance quality. In order to further understand the mechanism of lignin synthesis, transcriptome analysis was carried out on two cutting types of fruit samples from the slices treatment (P) and shreds treatment (S) stored for 24 h. Compared with the whole fruit (CK), 2281 and 2259 differentially expressed genes (EDGs) were identified in the slices and shreds treatments, respectively; 1442 up-regulated genes and 839 down-regulated genes were expressed as 2281 in the slices treatment; 1475 significantly up-regulated genes and 784 significantly down-regulated genes were expressed as 2259 in the shreds treatment; and 1582 DEGs were commonly detected between the slices treatment and shreds treatment, indicating that these DEGs were related to lignin synthesis. Gene Ontology (GO) enrichment and Kyoto Encyclopedia of Genes and Genomes (KEGG) pathway analysis showed that compared with the whole fruit, the metabolic pathways of amino acid metabolism, lipid metabolism, and secondary metabolism were affected by mechanical damages. This study revealed that JA biosynthesis was activated by mechanical damages, and the up-regulation of phenylalanine metabolism and phenylalanine, tyrosine, and tryptophan metabolism affected phenylpropanoid biosynthesis, which may promote lignin synthesis. Fifteen DEGs were selected for qRT-PCR validation, and the reliability and accuracy of transcriptome data were confirmed.

**Keywords:** fresh-cut cucumber; transcriptomic; lignin; amino acid metabolism; lipid metabolism; phenylpropanoid biosynthesis





## 1. Introduction

Globally, cucumber (*Cucumis sativus* L.) is one of the most popular Cucurbitaceae crops, with a global volume of production of more than 75.22 million tonnes [1]. Cucumbers are widely consumed because of their high nutritional values and beneficial secondary metabolites, including sugar, vitamins, carotene, and polyphenols, which can increase the antioxidant ability of vegetables [2]. Consumer demand for fresh-cut cucumber has substantially increased in recent years due to its freshness, good flavor, and convenience. However, fresh-cut cucumber is prone to problems such as accelerated physiological and biochemical reactions, water loss, microbial pollution, and lignification after mechanical damages, which seriously affect its taste and shelf life [3,4]. After the cucumber is cut, it activates its own defense system, such as the production of secondary metabolites, to resist mechanical damages. Among them, the lignin produced by the secondary metabolism will make the fresh-cut cucumber lignified, increasing its hardness, and whitening its cutting surface, thus seriously affecting its taste and appearance quality. Therefore, understanding

the biosynthesis mechanism of lignin, as well as delaying the occurrence of lignin, has become the focus of research in the fresh-cut fruit and vegetable industry.

As an important secondary metabolite, lignin is one of the components of plant cell walls, which can promote cell connection, maintain cell osmotic balance, and membrane integrity and reduce water loss, and the lignification of fresh-cut fruits and vegetables is the result of the reverse accumulation of lignin in the cell wall [5,6]. Research has shown that the suberin formed by the lignification of potato wounds can effectively avoid the infection of microorganisms and reduce the rate of transpiration and water loss, but the quality fission of many fruits and vegetables is also the result of the reverse accumulation of lignification in the cell wall, such as loquat, bamboo shoots, and carrots [7–9]. The lignin synthesis pathway is mainly divided into three stages: the shikimate metabolism pathway, the phenylpropanoid biosynthesis, and the lignin-specific synthesis pathway [10]. The synthesis of lignin is closely related to phenylalanine ammonia-lyase (PAL), cinnamyl-alcohol dehydrogenase (CAD), peroxidase (POD), and other enzymes.

At present, the research on cucumbers mainly focuses on the fresh-keeping mechanism, microorganisms in the tissue, and the safety of fresh-cut cucumber, while the impact of mechanical damage treatments on the quality of fresh-cut cucumber is not clear, especially because the lignification of fresh-cut cucumber caused by mechanical damages needs further research [11,12]. However, lignin synthesis is a complex process, including changes in gene levels. In recent years, with the development of genome sequencing and genomics technology, transcriptome technology has been applied more and more widely in fruit biology research. For example, it has been used to reveal the aging and preservation mechanism of fruits and vegetables such as apples, carrots, and potatoes [13–15]. Based on all transcripts of organisms, transcriptomics screens key differential genes generated in biological changes and uses bioinformatics to analyze rich differential metabolic pathways. In this study, transcriptome analysis based on RNA-seq was used to screen the EDGs between the treated and untreated cucumbers, and we analyzed the KEGG metabolic pathway to find out the key DEGs and key metabolic pathways related to the lignin synthesis of fresh-cut cucumber and clarify its lignin biosynthesis mechanism. It provides a new perspective and theoretical basis for improving the potential nutritional value of cucumbers and developing effective preservation methods to maintain cucumber quality.

## 2. Materials and Methods

### 2.1. Plant Materials, Treatments and Storage Conditions

Cucumber (*Cucumis sativus* L.) variety "Mici" was harvested in July 2022 and transported to the laboratory within 2 h, then selected in uniform size for the study. The selected cucumber fruits were dipped in 0.02% sodium hypochlorite for 5 min, washed with distilled water, and then placed on a ventilator to dry naturally. The fruits were cut into slices with a 3.5 cm diameter and 0.5 cm thickness (P); shreds in the shape of a cuboid with 4 cm length, 0.5 cm width, and 0.5 cm height (S); and with whole fruits as the control (CK). Based on a previous method [16], the calculated wounding intensities were 1.8 and 4.0 $cm^2 \, g^{-1}$ for slices and shreds, respectively. All the plants were then packed in 15 cm $\times$ 10 cm $\times$ 4 cm polyethylene films and stored at 4 °C and for 3 d with 95% relative humidity. Fruit samples were collected daily, frozen with liquid nitrogen, and smashed with a frozen crusher to obtain plant powder.

### 2.2. Measurement of Enzyme Activity

Approximately 1 g frozen sample was used for the PAL enzyme assay according to the protocol of the phenylamine ammonia-lyse measurement kit (PAL, Suzhou Comin Biotechnology Co., Ltd., Suzhou, China). One unit of PAL activity was defined as the amount of enzyme required to increase the absorbance of 0.1 at 290 nm $\times$ $g^{-1}$ $\times$ $min^{-1}$. Approximately 1 g of each frozen sample was used for the CAD enzyme assay according to the protocol of the cinnamyl-alcohol dehydrogenase measurement kit (CAD, Suzhou Comin Biotechnology Co., Ltd., Suzhou, China). One unit of CAD activity was defined

as the amount of enzyme required to increase the absorbance of 1 nmol NADPH at 340 nm $\times$ g$^{-1}$ $\times$ min$^{-1}$. The POD activity was determined by a modified method. Cucumber powder was added to 10 mL of phosphate-buffered saline (PBS, pH 6.4) containing 0.5% PVPP to determine POD activity. POD activity was measured with 0.9 mL of 0.2% guaiacol and 1 mL of 0.3 H$_2$O$_2$. The absorbance of POD was measured at 470 nm for 1 min. The enzyme required to cause an increase in absorbance of 0.01 in one minute is defined as one unit of activity [17]. Three independent replicates were used for each treatment.

*2.3. Determination of Lignin Contents*

Cucumber samples were taken as the material, and the samples were dried at 80 °C to constant mass, crushed and passed through a 40-mesh sieve, and weighed as about 2 mg (recorded as W), and the lignin content was determined by the acetyl bromide method [18,19], referring to the instructions of the lignin content kit (MZS, Suzhou Comin Biotechnology Co., Ltd., China), calculated according to the following equation:

$$\text{Lignin (mg/g)} = (A - A' - 0.0068) \div 0.0347 \times V \times 10^{-3} \div W \times T \tag{1}$$

where A is the measurement tube OD280 reading, A$'$ is the blank tube OD280 reading, V is the total reaction volume, T is the dilution multiple, and W is the sample weight.

*2.4. Transcriptomic Analysis*

2.4.1. RNA Extraction

Three cucumber samples from each treatment were randomly taken, namely, CK-1, CK-2, CK-3, P-1, P-2, P-3, S-1, S2, and S-3, for transcriptome analysis. TRIzol (Thermo Fisher, 15596018, CA, USA) was used to isolate and purify the RNA of cucumber samples according to the operation scheme provided by the manufacturer. Then, the quantity and purity of total RNA were controlled by the micro-nucleic acid protein quantitative instrument Nano Drop ND-1000 (Nano Drop, Wilmington, DE, USA).

2.4.2. Construction of Library

Oligo (dT) magnetic beads were used to specifically capture the mRNA containing polyadenylate (Poly-A) through two rounds of purification. The captured mRNA was fragmented under high temperature using a magnesium ion interruption kit. Fragmented RNA was synthesized into cDNA under the action of reverse transcriptase. The second chain was digested with UDG enzyme, pre-denatured at 95 °C for 3 min by PCR, denatured at 98 °C for a total of 8 cycles for 15 s, annealed at 60 °C for 15 s, extended at 72 °C for 30 s, and finally extended at 72 °C for 5 min to form a library with a fragment size of 300 bp $\pm$ 50 bp.

2.4.3. Sequencing Method

Illumina Novaseq TM 6000 (LC Bio Technology Co., Ltd., Hangzhou, China) was used to carry out two-terminal sequencing according to the standard operation, and the sequencing mode was PE150.

2.4.4. Verification of DEGs by qRT-PCR

The extraction of RNA in the sample was carried out according to the instructions of the plant RNA extraction kit. The reverse transcription steps of RNA refer to the instructions of the reverse transcription kit to reverse transcribe the RNA extracted from cucumbers into cDNA. The real-time fluorescence quantitative polymerase chain reaction (qRT-PCR) reaction system consisted of 10 μL 2X SYBR® Green Pro Taq HS Premix II, 0.8 μL positive primer, 0.8 μL reverse primer, 2 μL cDNA solution, and 6.4 μL RNase free water, with a total volume of 20 μL. The final concentration was 1X SYBR® Green Pro Taq HS Premix II, and 0.4 μM positive and reverse primers. The reaction procedure of qRT-PCR was 95 °C for 30 s; 40 cycles of 95 °C denaturation for 5 s, and 60 °C annealing for 30 s. The relative

quantification of genes was carried out using the $2^{-\Delta\Delta CT}$ method. The experiment was repeated three times.

### 2.5. Statistical Analysis

Microsoft Excel 2019 was used for data statistics and standard deviation calculation. SPSS 22.0 software was used for the Duncan difference significance test, and $p < 0.05$ indicated a significant difference. Figures were drawn with the omicstudio drawing module.

## 3. Results

### 3.1. Lignin Contents and Enzyme Activity of PAL, CAD, and POD

At earlier storage times, the lignin content increased in both cutting types and was the highest in the P treatment, indicating the P treatment had a lower wounding intensity than the S treatment (Figure 1a). Phenylpropane metabolism is the main pathway of lignin synthesis, and in order to illuminate the synthesis of lignin, the activities of critical enzymes PAL, CAD, and POD in the phenylpropane metabolic pathway were determined. PAL activity of the tissues increased slowly in the two cutting types and the control group (Figure 1b). The CAD activity decreased sharply during the first two days. At 2 d, CAD activity decreased by 46.0% and 53.6% in the P treatment and the S treatment, respectively, compared with the initial value. However, during the whole storage period, the CAD activity of the slices and shreds treatment was higher than that of the whole cucumbers (Figure 1c). POD activity increased during the 0–3 d of storage in all treatments. At 3 d, the POD activity in the P treatment and the S treatment was increased by 1.62 and 1.53 times, respectively, compared with the whole cucumbers (Figure 1d).

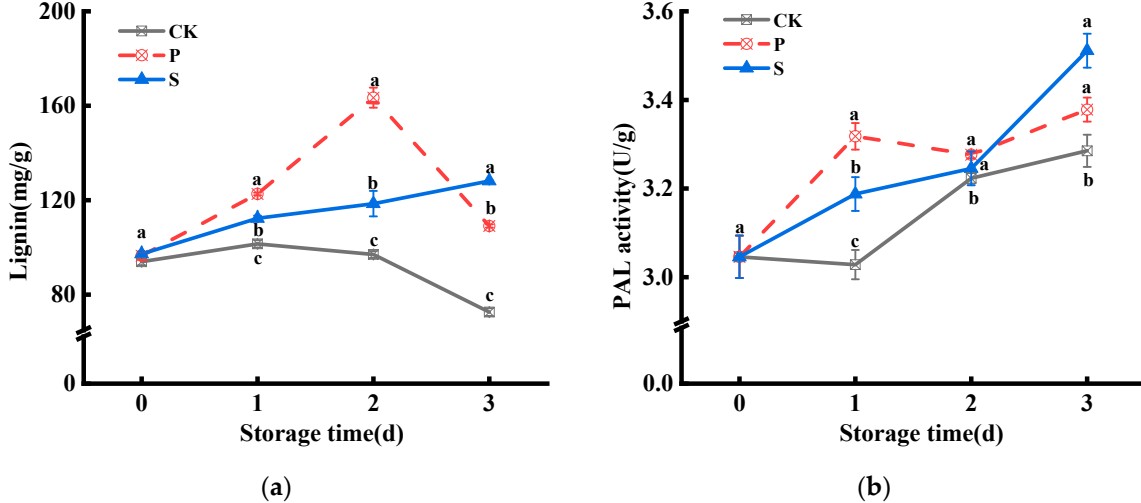

**Figure 1.** *Cont.*

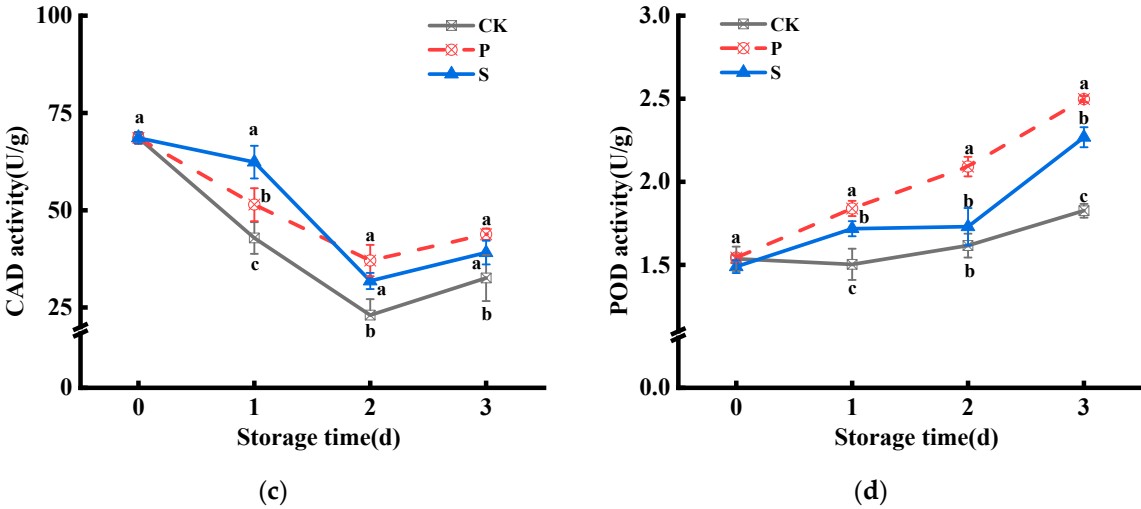

(**c**)                                                    (**d**)

**Figure 1.** Effect of cutting types on lignin content (**a**), PAL activity (**b**), CAD activity (**c**), and POD activity (**d**) in cucumbers. Columns with vertical bars represent the mean ± SD (n = 3). Different letters indicate significant differences between samples of different treatment groups for the same storage time ($p < 0.05$). CK represents the samples in the control treatment, P represents the samples in the slices treatment, and S represents the samples in the shreds treatment.

### 3.2. Transcriptomic Analysis

In order to ensure accurate and reliable analysis results, it is generally necessary to pre-process the original data (Table 1). After removing the sequencing connector (introduced during database construction) and low-quality sequencing data (caused by the error of the sequencer itself), we obtained 54.79 GB of effective data in total. The proportion of effective reads of each sample was more than 93.02%. The range distribution of Q20 (sequencing error rate is less than 0.01) was 99.95~99.96%, and that of Q30 (sequencing error rate is less than 0.001) was 96.58~96.94%. The GC content was 42.50%. The quality of sequencing data was good and suitable for further analysis.

**Table 1.** Statistical analysis of RNA-seq.

| Sample | Raw Data | | Valid Data | | Valid Ratio (%) | Q20% | Q30% | GC% |
|---|---|---|---|---|---|---|---|---|
| | Read | Base | Read | Base | | | | |
| CK | 40,771,589 | 6.11 G | 37,754,237 | 5.68 G | 93.02 | 99.96 | 96.80 | 42.50 |
| P | 39,870,298 | 5.98 G | 39,146,899 | 5.54 G | 93.11 | 99.96 | 96.74 | 42.50 |
| S | 41,079,656 | 6.03 G | 38,250,758 | 5.73 G | 93.11 | 99.96 | 96.67 | 42.50 |

CK represents the samples stored for the first day in the control group, P represents the samples stored for the first day in the slices treatment, and S represents the samples stored for the first day in the shreds treatment. Each group has three biological parallels.

Based on the FPKM value, Pearson correlation analysis was performed. The larger the correlation coefficient between samples, the better the clustering effects of samples (Figure 2a). The Pearson correlation coefficient $R^2$ of the three biological repeats in each sample group was greater than 0.999, indicating that the three biological repeats in the sequencing data of this transcriptome were reliable. The $R^2$ value between CK and S was the lowest, ranging from 0.906 to 0.914. The $R^2$ value between CK and P took second place, namely 0.93 to 0.931. The $R^2$ value between P and S was the largest, ranging from 0.983 to 0.987. Further, the principal component analysis (PCA) was carried out for the gene expression of all samples (Figure 2b), the three biological replicates of each group of samples were gathered together in space and there was a significant spatial distance between the sample groups, indicating that the samples in the same group had strong consistency and there were differences between groups, that is, there were differences

between samples processed by different cutting types. Therefore, it can be used for DEGs analysis.

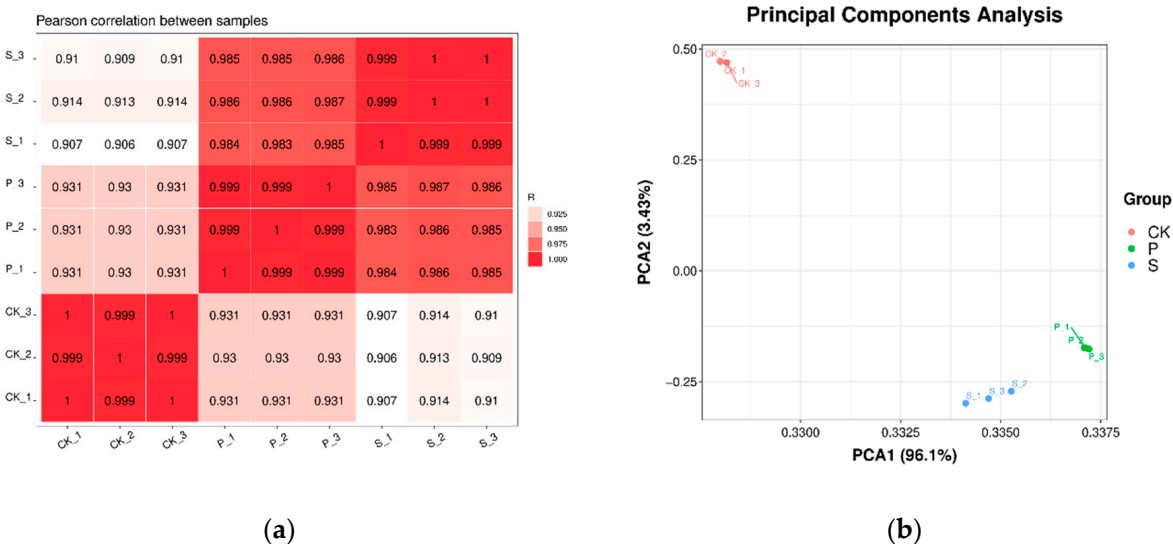

(**a**)　　　　　　　　　　　　　　　(**b**)

**Figure 2.** Pearson correlation analysis (**a**) and principal component analysis (**b**) were performed on the gene expression of all samples.

FC > 2 (up-regulated) or FC < 0.5 (down-regulated) and *p* < 0.05 are used as the threshold for screening DEGs. Volcanic maps of all genes in the differential expression analysis. With log2 (FC) as the abscissa and log10 (q value) as the ordinate, the DEGs among the CK, S, and P treatments after 24 h storage were shown (Figure 3a). Compared with CK, there were 2281 DEGs (1442 up-regulated genes and 839 down-regulated genes) in the P treatment, and a total of 2259 DEGs (1475 up-regulated genes and 784 down-regulated genes) were found in the S treatment. Moreover, there were 1582 common DEGs between P vs. CK and S vs. CK, of which the number of up-regulated genes was greater than that of down-regulated ones, indicating that more up-regulated genes were induced by mechanical damages in cucumbers, and they accelerated the expression of more genes involved in lignin biosynthesis in cucumbers (Figure 3b).

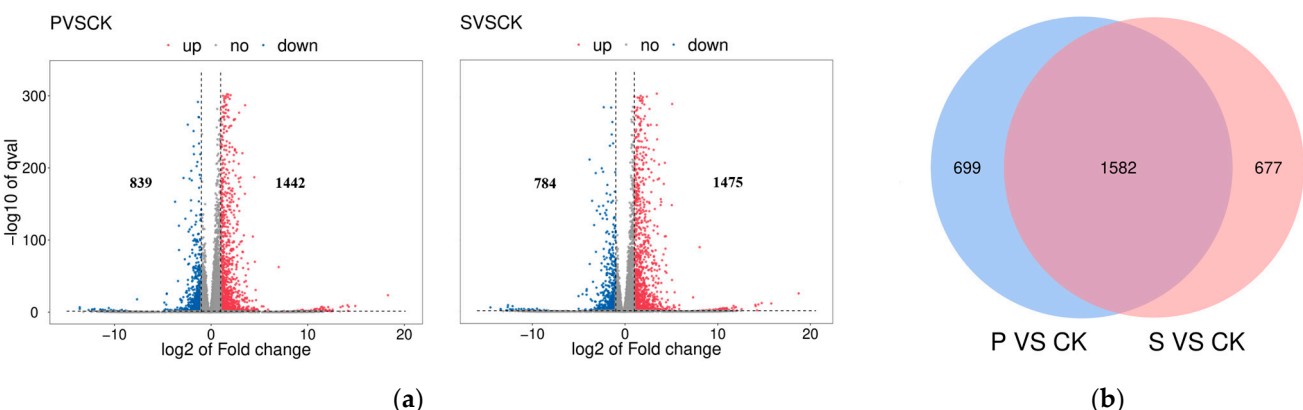

(**a**)　　　　　　　　　　　　　　　(**b**)

**Figure 3.** Volcanic map (**a**) and Venn diagram (**b**) of differentially expressed genes between P vs. CK and S vs. CK. The abscissa represents the change of differentially expressed genes multiple in different samples, the ordinate represents the statistical significance of the difference in gene expression.

### 3.3. GO Annotation of Differentially Expressed Genes

In P vs. CK and S vs. CK, 1348 DEGs (920 up-regulated and 518 down-regulated) and 694 DEGs (472 up-regulated and 222 down-regulated) were annotated, respectively. The GO functions are mainly divided into three categories: biological process (BP), cellular component (CC), and molecular function (MF). In P vs. CK, DEGs were enriched in 584 GO entries, of which 64 were significantly enriched ($p < 0.05$). In S vs. CK, the DEGs were enriched in 561 GO entries, of which 75 were significantly enriched ($p < 0.05$). When comparing the GO term in P vs. CK and S vs. CK, 59 out of 197 GO terms were found in common (Figure 4). DEGs involved in BP were mainly concentrated in DNA-templated, protein phosphorylation, oxidation–reduction process, transmembrane process, carbohydrate metabolic process, and lipid metabolic process. DEGs involved in CC were transcription factor complex, an integral component of the membrane and nucleus, and DEGs involved in MF were mainly ATP binding, protein binding, DNA binding, protein kinase activity, DNA-binding transcription factor activity, heme binding, and sequence-specific DNA binding.

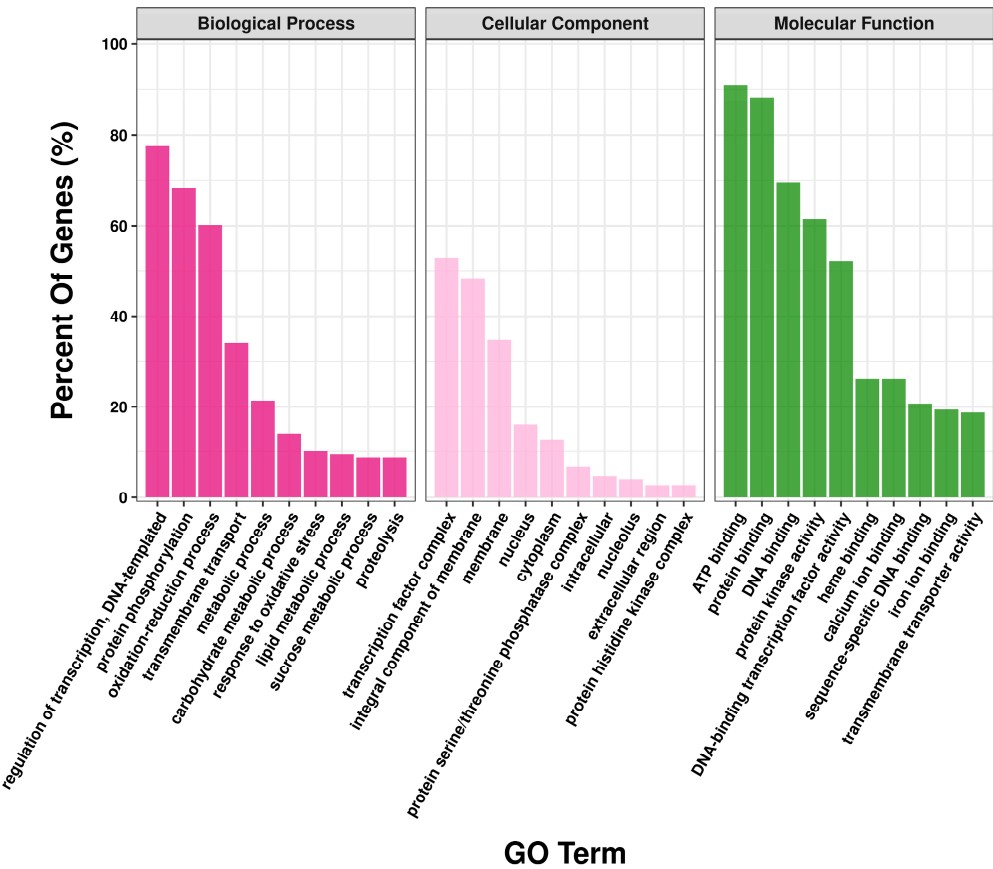

**Figure 4.** Main class of GO term annotated to differentially expressed genes in P vs. CK and S vs. CK. This analysis presented the top 10 pathways for each of BP, CC and MF.

### 3.4. KEGG Pathway Analysis

In order to characterize the pathways that were related to lignin biosynthesis by mechanical damages, we performed the metabolic pathways analysis on DEGs based on the KEGG database with $p < 0.05$ as the threshold. We found that 116 and 125 KEGG pathways were annotated by the DEGs of P vs. CK and S vs. CK, respectively. When comparing the KEGG pathway in P vs. CK and S vs. CK, 15 out of 94 KEGG pathways were found in common (Figure 5). The result showed that there were 10 pathways of metabolism, mainly including carbohydrate metabolism, energy metabolism, amino acid metabolism, biosynthesis of other secondary metabolites, lipid metabolism, nucleotide metabolism, metabolism

of terpenoids and polyketides, metabolism of cofactors and vitamins, metabolism of other amino acids, and glycan biosynthesis and metabolism.

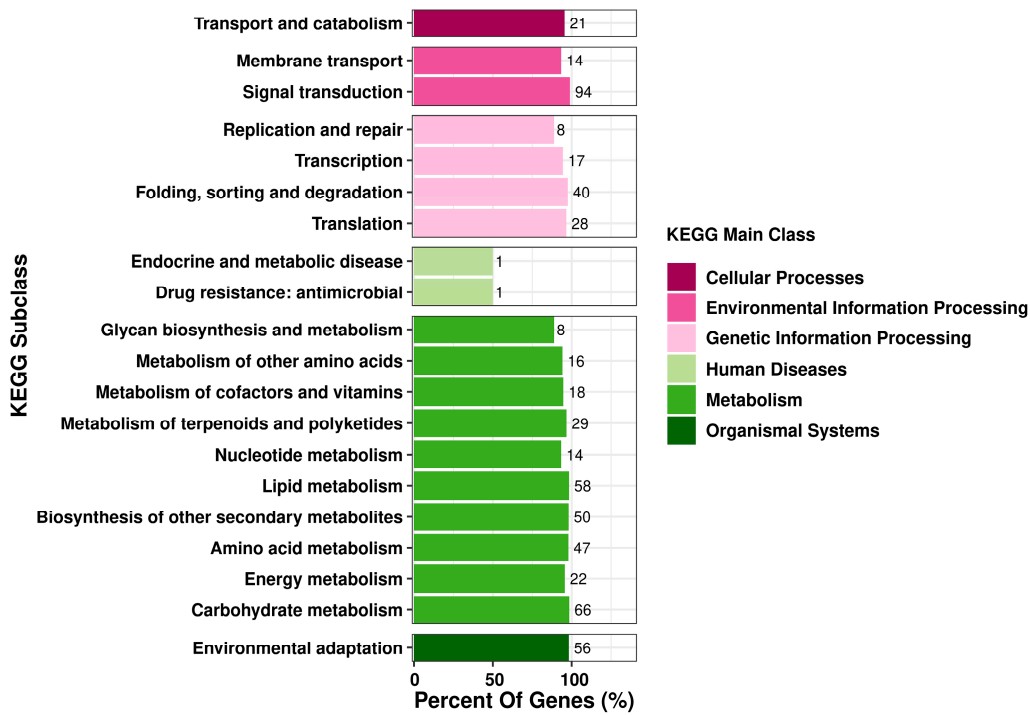

**Figure 5.** Main class of KEGG pathways annotated to differentially expressed genes in P vs. CK and S vs. CK. This analysis presented the top 20 pathways.

### 3.5. Critical Metabolic Pathways Induced in Response to Mechanical Damages in Cucumbers

Based on previous studies, we focused on three metabolism pathways, specifically lipid metabolism, amino acid metabolism, and biosynthesis of other secondary metabolites, which could be involved in mechanical damages [20,21]. It is very necessary to explore the interrelation and regulation of these metabolic pathways and find the key connection points between them for analyzing the mechanism of lignin biosynthesis in fresh-cut cucumber.

#### 3.5.1. Lipid Metabolism

In order to analyze the effect of the mechanical damages on the metabolism of cucumber lipid compounds, the statistical analysis of the DEGs in the P treatment and the S treatment is shown (Table 2). There were 100 DEGs (88 were significantly up-regulated and 12 were significantly down-regulated) and 98 DEGs (84 were significantly up-regulated and 14 were significantly down-regulated) involved in lipid metabolism in the slices and shreds cucumbers. Among the lipid metabolism pathways annotated by DEGs in the two cutting type groups, linoleic acid metabolism ($p = 0.0023$) was significantly enriched in the slices group and fatty acid elongation ($p = 0.0350$) in the shreds one. However, the pathways with significant differences were alpha-linolenic acid metabolism ($p = 0.0001$), and the number of DEGs was 20 in both of the cutting types, in which there were 19 up-regulated and 1 down-regulated DEGs in the slices group and 18 up-regulated and 2 down-regulated in the shreds one, respectively (Figure 6). There were four *PLA1*, six *13S LOX2*, three *AOS*, two *AOC*, and one *JOMJ* up-regulated both in P vs. CK and S vs. CK. Compared with CK, the *HPL1* was down-regulated in the P treatment and not significantly expressed in the S treatment. One *OPR* was down-regulated in the S treatment and not significantly expressed in the P treatment.

**Table 2.** Lipid metabolism in fresh-cut cucumber.

| Lipid Metabolism | P vs. CK | | S vs. CK | |
|---|---|---|---|---|
| | DEGs | *p* | DEGs | *p* |
| Alpha-linolenic acid metabolism | 20 | 0.0001 | 20 | 0.0001 |
| Fatty acid elongation | 5 | 0.2169 | 7 | 0.0350 |
| Linoleic acid metabolism | 10 | 0.0023 | 7 | 0.0615 |
| Ether lipid metabolism | 7 | 0.1773 | 8 | 0.0821 |
| Secondary bile acid biosynthesis | 1 | 0.1918 | 1 | 0.1894 |
| Glycerophospholipid metabolism | 14 | 0.2693 | 14 | 0.2536 |
| Arachidonic acid metabolism | 5 | 0.1801 | 4 | 0.3503 |
| Glycerolipid metabolism | 13 | 0.3768 | 13 | 0.3589 |
| Biosynthesis of unsaturated fatty acids | 4 | 0.7434 | 5 | 0.5479 |
| Cutin, suberine, and wax biosynthesis | 7 | 0.1255 | 4 | 0.6144 |
| Sphingolipid metabolism | 9 | 0.4929 | 8 | 0.6222 |
| Steroid biosynthesis | 1 | 0.9634 | 2 | 0.8297 |
| Fatty acid degradation | 3 | 0.9573 | 4 | 0.8786 |
| Fatty acid biosynthesis | 1 | 0.9961 | 1 | 0.9958 |

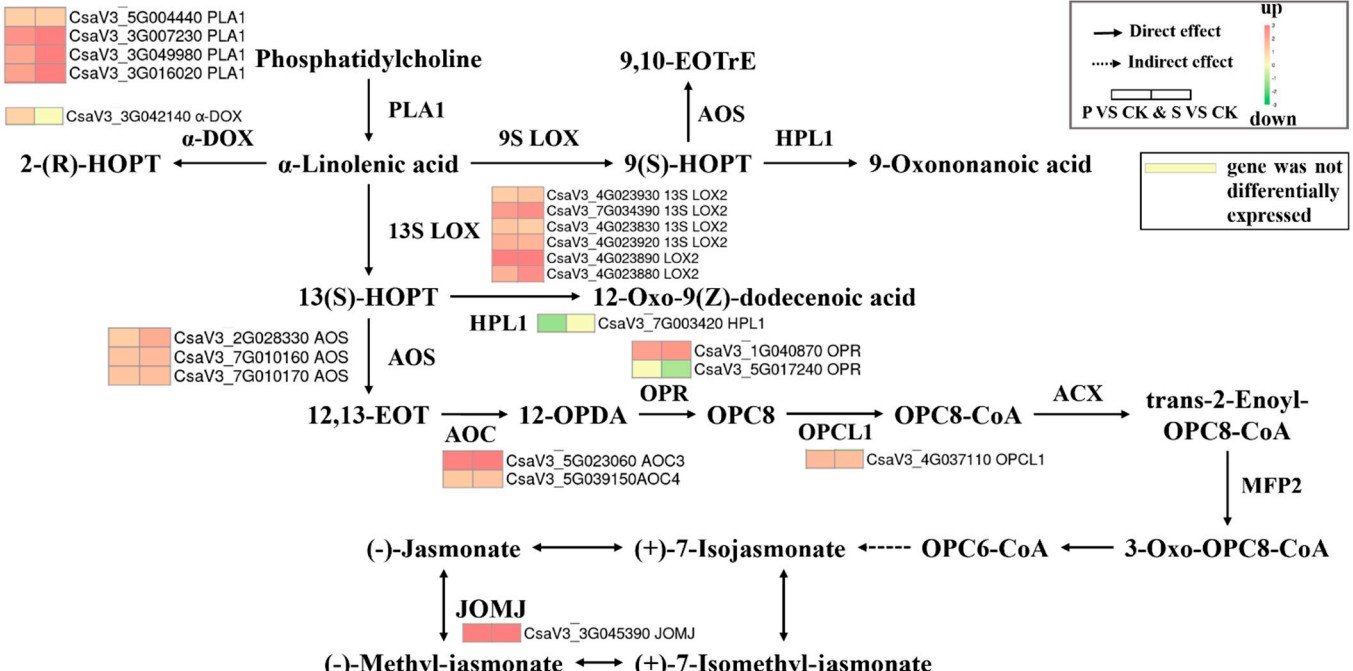

**Figure 6.** Changes in the alpha-linolenic acid metabolism pathway in fresh-cut cucumber. Genes with differential abundance involved in alpha-linolenic acid metabolism are mapped to the corresponding pathway. Red indicates up-regulation, and green indicates down-regulation. PLA: peroxisomal acyl-coenzyme A oxidase; 13S LOX: linoleate 13S-lipoxygenase; AOS: allene oxide synthase; AOC: allene oxide cyclase; a-DOX: alpha-dioxygenase; HPL: linolenate hydroperoxide lyase; OPR: 12-oxophytodienoate reductase; OPCL: OPC-8:0 CoA ligase; JOMJ: jasmonate O-methyltransferase.

### 3.5.2. Amino Acid Metabolism

In order to analyze the effect of mechanical damages of cucumbers on amino acid metabolism, the DEGs of P vs. CK and S vs. CK were statistically analyzed (Table 3). There are 78 DEGs in P vs. CK, of which 62 are up-regulated and 16 are down-regulated. These DEGs participate in 11 amino acid metabolism pathways, of which three pathways are significantly different ($p < 0.05$), namely phenylalanine metabolism, phenylalanine, tyrosine and tryptophan biosynthesis, and arginine and proline metabolism. There are 80 DEGs in S vs. CK, of which 59 are up-regulated and 21 are down-regulated. These DEGs are mainly distributed in 14 amino acid metabolic pathways, of which three pathways are

significantly different ($p < 0.05$), namely phenylalanine metabolism, phenylalanine, tyrosine, and tryptophan biosynthesis as well as glycine, serine, and threonine metabolism. The significantly different pathways shared by DEGs in P vs. CK and S vs. CK are phenylalanine metabolism, phenylalanine, tyrosine, and tryptophan biosynthesis. Among them, the 16 DEGs in phenylalanine metabolism are all up-regulated in the two cutting types; 11 DEGs of slice treatment are all up-regulated, followed by 10 DEGs (8 up-regulated and 2 down-regulated) in the shreds treatment of phenylalanine, tyrosine and tryptophan biosynthesis (Figure 7). By analyzing the genes in these two metabolic pathways, *SK*, *ADT*, *POA*, and *PAL* were found significantly up-regulated in the two cutting type treatments: *TSase A* and *ADH* were significantly down-regulated in the shreds cucumbers, but different from the slices treatment group that *CS*, *CM*, and *PPA-AT* in the shreds one did not change significantly.

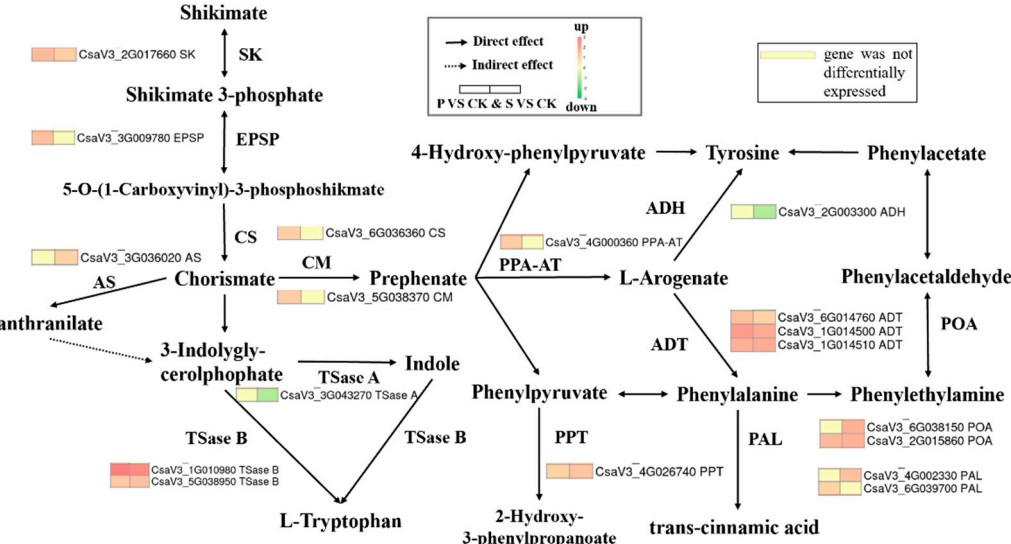

**Figure 7.** Changes in the phenylalanine, tyrosine, and tryptophan biosynthesis and phenylalanine metabolism pathway in fresh-cut cucumber. Genes with differential abundance involved in phenylalanine, tyrosine and tryptophan biosynthesis and phenylalanine metabolism are mapped to the corresponding pathway. Red indicates up-regulation, and green indicates down-regulation. SK: Shikimate kinase; EPSP: 3-phosphoshikimate 1-carboxyvinyltransferase; CS: chorismate synthase; CM: chorismate mutase; TSase A: tryptophan synthase alpha chain; TSase B: tryptophan synthase beta chain; ADH: arogenate dehydrogenase; ADT: arogenate dehydratase; PPA-AT: prephenate aminotransferase; PPT: phenylpyruvate tautomerase; POA: primary amine oxidase; PAL: phenylalanine ammonia-lyase; AS: anthranilate synthase.

**Table 3.** Amino acids metabolism in fresh-cut cucumber.

| Amino Acid Metabolism | P vs. CK | | S vs. CK | |
| --- | --- | --- | --- | --- |
| | DEGs | *p* | DEGs | *p* |
| Phenylalanine metabolism | 16 | 0.0028 | 16 | 0.0025 |
| Phenylalanine, tyrosine, and tryptophan biosynthesis | 11 | 0.0098 | 10 | 0.0236 |
| Glycine, serine, and threonine metabolism | 13 | 0.1175 | 15 | 0.0316 |
| Alanine, aspartate and glutamate metabolism | 7 | 0.1186 | 6 | 0.5433 |
| Arginine biosynthesis | 7 | 0.1773 | 4 | 0.6864 |
| Valine, leucine, and isoleucine biosynthesis | 2 | 0.7133 | 2 | 0.7063 |
| Lysine biosynthesis | 1 | 0.7497 | 1 | 0.7450 |
| Tryptophan metabolism | 3 | 0.9007 | 4 | 0.7623 |
| Tyrosine metabolism | 4 | 0.797 | 4 | 0.7884 |
| Cysteine and methionine metabolism | 11 | 0.7487 | 10 | 0.8269 |

**Table 3.** *Cont.*

| Amino Acid Metabolism | P vs. CK | | S vs. CK | |
|---|---|---|---|---|
| | DEGs | *p* | DEGs | *p* |
| Histidine metabolism | - | - | 1 | 0.8493 |
| Valine, leucine, and isoleucine degradation | - | - | 3 | 0.9717 |
| Arginine and proline metabolism | 3 | 0.0423 | 3 | 0.9777 |
| Lysine degradation | - | - | 1 | 0.9852 |

- indicates that it is not checked out.

### 3.5.3. Phenylpropanoid Biosynthesis

There were 55 DEGs involved in the phenylpropanoid biosynthesis pathway in P vs. CK (40 up-regulated genes and 15 down-regulated genes, and 53 DEGs (37 up-regulated genes and 16 down-regulated genes) were found in S vs. CK. Most genes were up-regulated in the phenylpropanoid biosynthesis pathway, indicating that mechanical damages could promote lignin biosynthesis (Figure 8). The mechanical damages induced seven *PAL*, two *CYP37A*, three *4CL*, and one *CAD* up-regulation in the two cutting types of cucumbers. Among the DEGs involved in the phenylpropanoid biosynthesis pathway in cucumbers subjected to mechanical damages, the changes of *CCR* and *POD* family genes were the most complex. Compared with the control group, three *CCR2* were up-regulated in the slices and shreds groups, five *CCR1* (three *CCR1* up-regulated, two *CCR1* down-regulated) in slices, and three *CCR1* (two *CCR1* up-regulated, one *CCR1* down-regulated) in shreds. Three lignin-forming *POD* were up-regulated and three *POD17* were down-regulated both in the slices and shreds cucumbers, two *POD5* in the slices cucumbers were significantly up-regulated, while no significant changes in the shreds treatment, one *POD49* was only up-regulated in the shreds cucumbers, and one *POD4* was only up-regulated in the slices one.

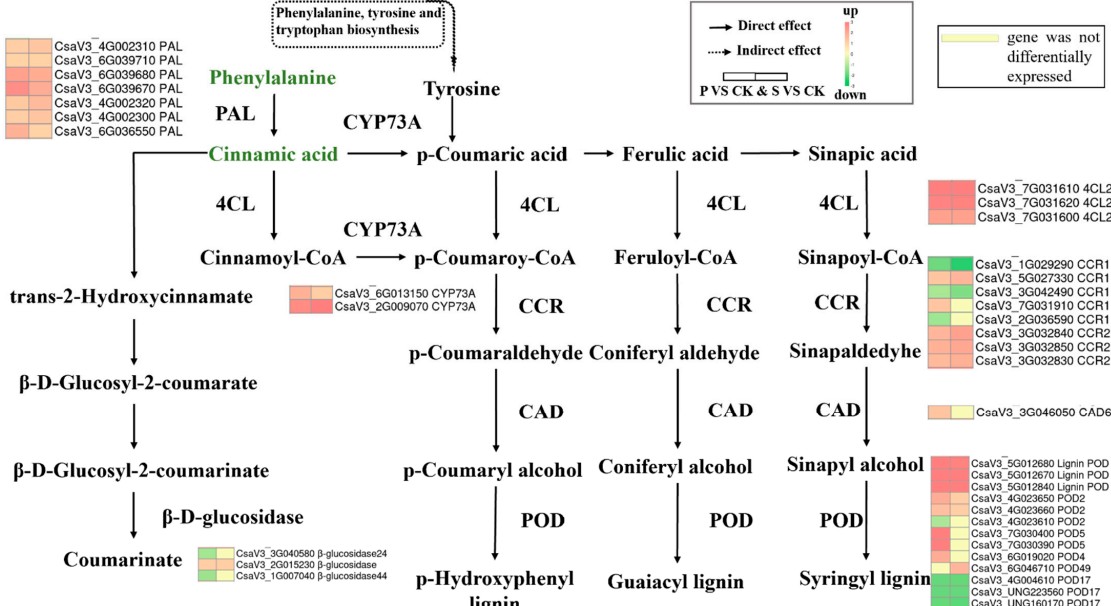

**Figure 8.** Changes in the Phenylpropanoid biosynthesis pathway in fresh-cut cucumber. Genes with differential abundance involved in phenylpropanoid biosynthesis metabolism are mapped to the corresponding pathway. Red indicates up-regulation, and green indicates down-regulation. PAL: phenylalanine ammonia-lyase; CYP73A: trans-cinnamate 4-monooxygenase; Lignin-forming POD: lignin-forming anionic peroxidase; POD: peroxidase; CCR: cinnamoyl-CoA reductase; 4CL: 4-coumarate-CoA ligase; β-glucosidase: beta-glucosidase; CAD: cinnamyl-alcohol dehydrogenase.

### 3.6. Verification of DEGs by qRT-PCR

A total of 15 DEGs were selected to validate the RNA-seq data by qRT-PCR evaluation using specific primers (Table S1). Relative expression levels of 15 selected genes were analyzed by qRT-PCR (Figure 9a), and the square value of the correlation coefficient R in the linear regression analysis was 0.925 (Figure 9b), indicating that the transcriptome results were reliable in our study.

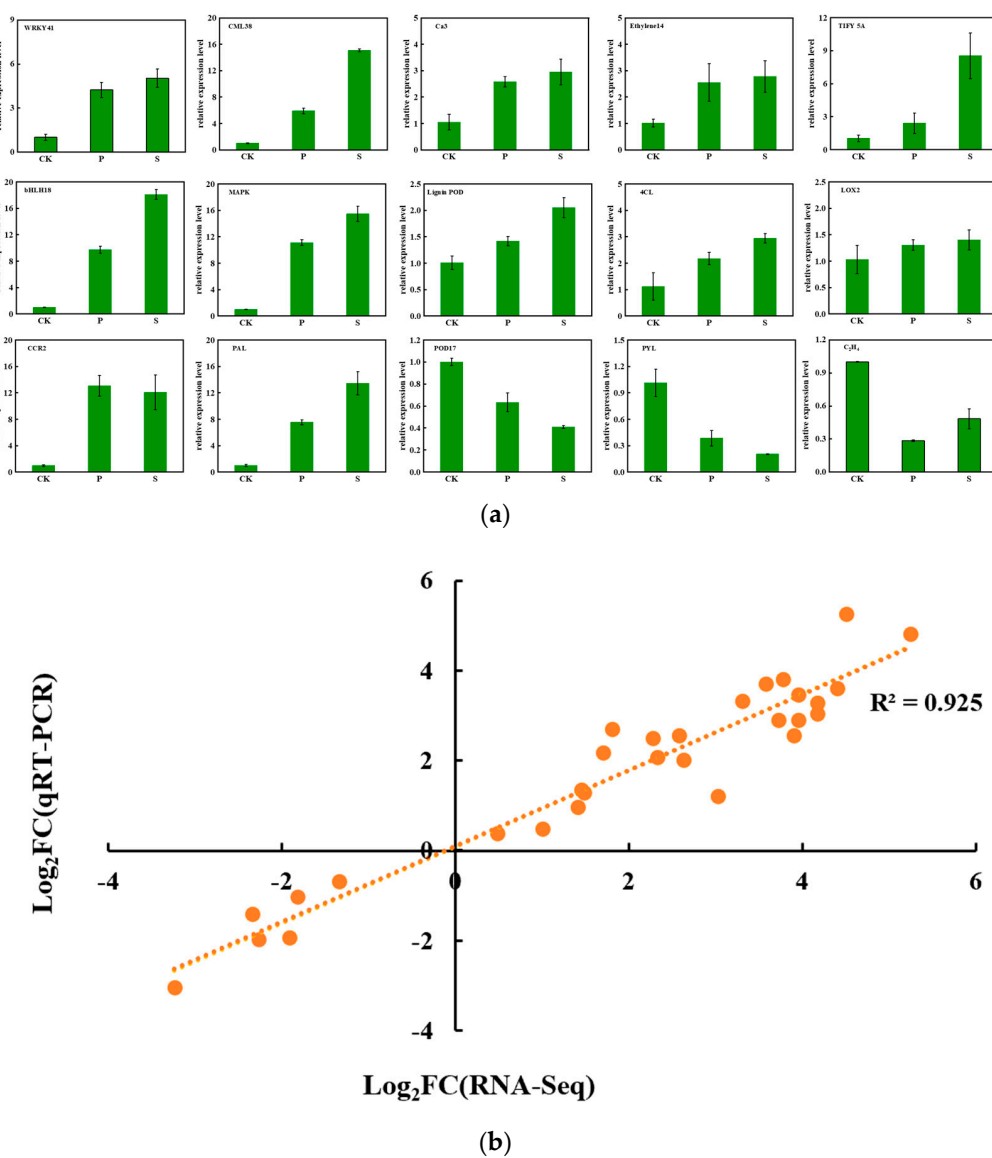

(**a**)

(**b**)

**Figure 9.** Validation of RNA-seq results. Relative expression levels of 15 selected genes were analyzed by qRT-PCR (**a**). Regression analysis of gene expression levels based on log2 FC (RNA-seq) (*x*-axis) and log2 FC (qRT-PCR) (*y*-axis) (**b**).

## 4. Discussion

The lignification of fruits will inevitably affect the quality of vegetables and fruits, and fresh-cut cucumber is prone to lignification during storage, which increases its lignin content and hardness, leading to a reduction in consumer acceptance. The physiological study found that the two cutting types could make lignin accumulate in a large amount, and mechanical damages could promote the activities of PAL and POD. Therefore, we further analyzed the mechanism of lignin biosynthesis in transcriptomics. The amino acid metabolism, the lipid acid metabolism, and the biosynthesis of other secondary metabolites

were the enriched pathways, regardless of the cutting styles, indicating that these processes were involved in the response of the lignin biosynthesis of cucumbers to mechanical damages. Thus, the analysis of the possible functions of DEGs by comparison between them would help to reveal the mechanism of lignin biosynthesis in fresh-cut cucumber. The DEGs were significantly enriched in alpha-linoleic acid metabolism, phenylpropane metabolism, phenylalanine, tyrosine and tryptophan biosynthesis, and phenylpropanoid biosynthesis by the analysis of KEGG enrichment. The results suggested that these pathways might be closely related to lignin synthesis induced by mechanical damages.

Mechanical damages can make fruits and vegetables produce JA signaling, which triggers a series of reactions and secondary metabolites that are synthesized for repairing the damages of tissue [22,23]. Previous reports have shown that JA are important plant hormones with multiple roles in mechanical damages, which can activate secondary metabolite biosynthesis pathways in the defense response, especially in lignin biosynthesis or development pathways [24]. Transcriptomic analysis showed that the *LOX* in cucumbers was significantly up-regulated by mechanical damage treatment. The main function of LOX was to oxidize linolenic acid by inserting oxygen at the C-13 position, providing a material precursor for JA synthesis. Secondly, *AOS* and *AOC* were significantly up-regulated, both of which could catalyze the conversion of alkoxides to OPDA, and finally, JA was generated under the catalysis of JOMJ [25]. Therefore, mechanical damages can induce JA synthesis, which may be an important reason for accelerating the lignin synthesis of fresh-cut cucumber (Figure 6).

The common function of amino acids is used as raw materials for protein synthesis. They can also be used as precursors for the synthesis of vitamins, nucleotides, terpenes, alkaloids, phenolic acids, flavonoids, lignin, purines, and pyrimidines [26,27]. Phenylalanine (Phe), tyrosine (Tyr), and tryptophan (Try) are aromatic amino acids in plants, which are synthesized mainly through the shikimate pathway. Phosphoenolpyruvate (PEP) produced by the glycolysis pathway and erythrose 4-phosphate (E4P) produced by the pentose phosphate pathway enter the shikimate metabolism pathway under the catalysis of enzymes [28,29]. SK can convert shikimate into shikimate 3-phosphate, which can be further converted into chorismate [30]. Under the catalysis of CM, prephenate is generated and aromatic amino acids are further synthesized. The reaction is divided into two ways, one of which is converted to Phe and Tyr, and the other way is to generate Try (Figure 7). It can be seen that although the different cutting type treatments have a different impact on the phenylalanine, tyrosine, and tryptophan biosynthesis in cucumbers, they all accelerate the conversion of shikimate to Phe, which can provide more carbon sources for lignin synthesis and then accelerate the synthesis of lignin. Phe is an essential amino acid for plants, which can be used as a carbon source and affect the growth and development of plants by participating in protein synthesis [31]. At the same time, in the secondary metabolism of plant cells, Phe is an important precursor for the synthesis of lignin, alkaloids, flavonoids, and other secondary substances, so it has an important physiological significance [32]. Phenylalanine metabolism is the main way to determine the fate of phenylalanine, and also an important intermediate way to connect primary metabolism (such as TCA cycle and glycolysis) and secondary metabolism (such as phenylpropanoid biosynthesis). Therefore, mechanical damages can promote the accumulation of Phe and provide more precursors for lignin synthesis.

Lignin is an important final metabolite of cucumbers, which is mainly synthesized through the phenylpropane metabolic pathway [33,34]. Cutting treatment has significant effects on genes in the phenylpropane metabolism pathway of cucumbers. The mechanical damages induced PAL up-regulation both in P vs. CK and S vs. CK (Figure 9). As the first key enzyme in lignin synthesis, PAL could catalyze phenylalanine to produce cinnamic acid, providing more precursors for the lignin synthesis pathway of phenylpropanoid biosynthesis. This result was similar to previous reports in fresh-cut broccoli and carrot [35,36]. In this study, some novel information related to phenylpropanoid biosynthesis was found in fresh-cut cucumbers, and *4CL*, *CCR*, *CAD*, and *POD* were identified as

DEGs in both P and S treatments. These genes were the critical genes of the lignin synthesis pathway. *CCR*, *CAD*, and *POD* are the key rate-limiting enzymes for lignin synthesis. It is worth mentioning that the effect of mechanical damages on CCR family genes was complicated. The *CCR* family genes were simply divided into *CCR1* and *CCR2*, in which *CCR2* was all up-regulated in the two cutting types of cucumbers; however, the changes of *CCR1* were complex, and some were up-regulated while others were down-regulated. The main reason may be that *CCR* catalyzes the reduction of hydroxycinnamoyl-CoA to corresponding cinnamaldehydes, thereby further regulating the synthesis process of precursor substances for lignin. *CAD* can catalyze the synthesis of mono-lignin in the process of lignin synthesis, reduce cinnamaldehyde to cinnamyl-alcohol, and provide precursors for lignin synthesis. The results showed that the fresh-cut treatment promoted the lignin synthesis pathway, and the previous research results of potato and kiwifruit [37,38] also showed that the fresh-cut treatment accelerated the lignin synthesis, which was consistent with the results of this study. However, the *POD* family has multiple functions in the process of plant metabolism. On the one hand, POD, with the dual functions of clearing active oxygen and accelerating browning, can catalyze hydrogen peroxide and oxidize phenols to form brown products in the presence of $H_2O_2$. On the other hand, *POD*, which can catalyze the polymerization of lignin monomers to produce lignin, participates in the final catalytic reaction of lignin formation in the downstream pathway of phenylpropanoid biosynthesis. In this study, the mechanical damage treatment increased the expression of three lignin-forming *POD* all up-regulated in the two cutting types of cucumbers, which can catalyze the polymerization of lignin monomers to complete the process of lignin, the last step of lignin biosynthesis. Taken together, the results suggested that the lignin biosynthesis genes may be up-regulated by mechanical damages according to the phenylpropanoid metabolism, resulting in a promotion of lignification in fresh-cut cucumber.

## 5. Conclusions

The gene profile of fresh-cut cucumber subjected to mechanical damages was obtained through the transcriptome method, P vs. CK involved 2281 (1442 up-regulated and 839 down-regulated genes) differentially expressed genes, and S vs. CK involved 2259 (1475 up-regulated and 784 down-regulated genes) differentially expressed genes. Bioinformatics analyses revealed that DEGs are mainly involved in alpha-linolenic acid metabolism and phenylalanine metabolism, as well as phenylalanine, tyrosine, and tryptophan biosynthesis. Meanwhile, phenylpropanoid biosynthesis was activated by mechanical damages by regulating some critical enzymes such as PAL, CAD, and POD. Moreover, this study revealed that JA signals were activated by inducing the up-regulated expression of genes such as *LOX*, *AOS*, and *AOC* by mechanical damages, and the JA signals significantly up-regulated the expression of genes such as *PAL*, *C4H*, *4CL*, *CCR*, and *CAD*, which accelerated the synthesis of lignin monomers and finally up-regulated the expression of lignin-forming *POD*, which accelerated the synthesis of lignin. This investigation provides useful information to better understand the molecular mechanisms of mechanical damages in post-harvest cucumbers and then further enhance the potential nutritional values in the processing of harvested cucumbers.

**Supplementary Materials:** The following supporting information can be downloaded at: https://www.mdpi.com/article/10.3390/horticulturae9040500/s1, Table S1: Primers used in qRT-PCR.

**Author Contributions:** Conceptualization, W.H. and Y.W.; methodology, W.H., C.C. and Y.W.; software, W.H., C.C. and Y.W.; validation, C.C., Y.W. and N.Y.; formal analysis, W.H., C.C. and Y.W.; investigation, W.H., C.C. and Y.W.; resources, W.H.; data curation, W.H. and Y.W.; writing—original draft preparation, W.H., C.C. and Y.W.; writing—review and editing, Y.W., N.Y. and Y.G.; visualization, Y.W., N.Y. and Y.G.; supervision, N.Y. and Y.G.; project administration, W.H.; funding acquisition, W.H. All authors have read and agreed to the published version of the manuscript.

**Funding:** This research was supported by the "Thirteenth Five-Year Plan" for the National Key Research and Development Program (No. 2016YFD0400903), National Natural Science Foundation of China (No. 31471923).

**Data Availability Statement:** The datasets used and/or analyzed during the current study are available from the corresponding author upon reasonable request.

**Conflicts of Interest:** The authors declare no conflict of interest.

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
