# Peer review of "Transcriptomic Analysis Reveals the Mechanism of Lignin Biosynthesis in Fresh-Cut Cucumber"

_horticulturae, doi:10.3390/horticulturae9040500_

Round 1
Reviewer 1 Report
The manuscript entitle 'Transcriptomic Analysis Reveals the Mechanism of Lignin Biosynthesis in Fresh-cut Cucumber' has been well written and some changes are required. i have appended all the comments in the attached PDF. Authors can take a line by line review of the manuscript for any English corrections.

Reviewer 2 Report
The manuscript by Wang et al. is devoted to transcriptomic analysis of the mechanism of lignin biosynthesis in fresh-cut cucumber. The work seems to be interesting; however, I suppose that perspectives of using these results for solution of agricultural tasks should be discussed in more details. For example, can these results be used for development of transgenic plants, which will have limited lignin production after mechanical damages?
Specific points:
1. Why was 3-day time interval with step 1 day used for investigation? Can time intervals with more duration be important for the fresh-cut cucumber storage? It should be clarified.
2. Figure 1a: Dynamic of lignin content in S-variant seems to be two-phases. What were mechanisms of this effect? Can this effect be important for fresh-cut cucumber storage?
3. Figure 1: Abbreviatures (CK, P, S) should be deciphered in caption of this figure.
Reviewer 3 Report
The article of the authors is devoted to a very important topic related to food security, which is very important with the ever-increasing population of the planet. The work was done at a good methodological level, but some points raise questions in my mind.
Firstly, the cucumbers were bought in the market - so you cannot be sure that they are fruits of the same age, that they were taken at the same time, that they are the same variety, etc. It is not clear how the authors can draw conclusions without knowing that the initial experimental material is homogeneous?
Secondly, the measurement of lignin on a spectrophotometer is not an accurate method. For reliable data, it is necessary to use GC / MS in its various versions (https://doi.org/10.1021/acs.analchem.7b02632, https://doi.org/10.1016/0165-2370(93)80041-W ). The authors do not even provide a single reference to the technique they used, which makes one even more concerned about the reliability of the method and the reliability of the data obtained.
Thirdly, in the statistics section, it is not indicated which post-hoc was used for statistical processing? And not a single figure contains information about what values differ reliably !!!
Fourthly, part of the captions in Figure 6, 7, 8, and the entire fragment A of Figure 9 is not readable. You need to make the signatures larger or clearer with an increase in the resolution of the picture.
Round 2
Reviewer 3 Report
Dear authors, you still haven't answered the question of which post-hoc was used? Duncan's test, HSD, NHSD or some other. Add this information to the methodology section.
